# Human ELISA Detects anti-SARS-CoV-2 Antibodies in Cats: Seroprevalence and Risk Factors for Virus Spread in Domestic and Stray Cats in Bulgaria

**DOI:** 10.3390/vetsci10010042

**Published:** 2023-01-06

**Authors:** Ivo Sirakov, Nikolina Rusenova, Anton Rusenov, Raina Gergova, Tanya Strateva

**Affiliations:** 1Department of Medical Microbiology, Medical Faculty, Medical University—Sofia, 1431 Sofia, Bulgaria; 2Department of Veterinary Microbiology, Infectious and Parasitic Diseases, Faculty of Veterinary Medicine, Trakia University, 6000 Stara Zagora, Bulgaria; 3Department of Internal Noninfectious Diseases, Faculty of Veterinary Medicine, Trakia University, 6000 Stara Zagora, Bulgaria

**Keywords:** ELISA, SARS-CoV-2, antibodies, humans, cats

## Abstract

**Simple Summary:**

The aim of the study was to verify whether the human DR-ELISA for the detection of antibodies against the agent of COVID-19 can be applied in cats, as well as to assess the risk factors that determine the spread of the virus among the cat population in Bulgaria. Therefore, 92 serum samples collected from 68 domestic and 24 stray cats were analyzed and compared with a multi-species ELISA kit. The results showed 83.33% positive results in stray cats and 41.18% in domestic cats, respectively, by both assays. Cats under 7 years had a five times higher risk than those over 7 years. The risk was seven times higher for stray cats than for domestic cats. Additionally, the results indicate the highest risk for cats in villages. This study demonstrates that human DR-ELISA may be helpful in monitoring the circulation of the virus in cats.

**Abstract:**

The aim of this study was to verify whether the human DR-ELISA for the detection of anti-SARS-CoV-2 antibodies can be applied in cats, and to assess the risk factors that determine the spread of the virus among the cat population in Bulgaria. The study included 92 serum samples collected from 68 domestic and 24 stray cats aged from 3 months to 20 years of age in the period of January–June 2021. The samples originated from three regions in Bulgaria and from three places of inhabitance. DR-ELISA based on peroxidase-labeled SARS-CoV-2 N protein was employed to detect IgA, IgG and IgM antibodies in the samples. Subsequently, the results were compared with a commercially available multi-species ELISA kit. There was high seroprevalence (83.33%) in stray cats and 41.18% in domestic cats, confirmed by the human and veterinary ELISA kit. The positive cases in the regional cities were 42.86%, in small towns 50% and in villages 78.26%. Cats under 7 years had a five times higher risk than those over 7 years (*p* = 0.001). The risk was seven times higher for stray cats than for domestic cats (*p* = 0.001). In addition, the results indicate that the risk was the highest for cats in villages (*p* = 0.006) compared to cats in other places of inhabitance. This study demonstrates that human DR-ELISA may be successfully applied to monitor the circulation of SARS-CoV-2 in cats and other susceptible species. Cats might serve as sentinel animals for tracking the virus in nature and in inhabited areas (strays) and to discover asymptomatic cases in humans/owners.

## 1. Introduction

The SARS-CoV-2 pandemic raised questions about the origin of the virus [1], its natural hosts [2] and its potential to act as a reservoir [3]. The animal species known to be highly susceptible to SARS-CoV-2 include cats, ferrets [2] and minks [4]. They often serve as laboratory animals in virus-related studies [5]. Unlike other animals susceptible to this virus, cats are a popular pet around the world. Thus, it is important to determine how significant cats are as a host or reservoir of SARS-CoV-2 [6]. This depends on the possibility of natural infection and subsequent circulation of the virus in the cat population.

That is why the attempts to control SARS-CoV-2 should focus on monitoring its spread not only in humans but also among other susceptible species, such as cats. The symptoms in cats are similar to other diseases caused by *Feline herpesvirus* (FHV), *Chlamydophila felis*, *Feline calicivirus* (FCV) and *Mycoplasma felis*. Therefore, it is recommended to include SARS-CoV-2 in the panel for differential diagnosis [7].

The molecular methods for diagnosis of SARS-CoV-2 that rely on the detection of nucleic acids are informative of ongoing infection, whereas the enzyme-linked immunosorbent assay (ELISA) for detection of antibodies against the virus can be useful both for control of vaccination in humans and animals, and for tracing the spread of the virus among susceptible species. At the beginning of the pandemic, there were no commercially available kits for the detection of SARS-CoV-2 in species other than humans. Therefore, researchers used a virus neutralization test (VNT), which requires a certain biosafety level of 3, and developed an indirect ELISA [8,9,10]. Then, studies reported two commercial kits based on a double antigen ELISA designed for different animal species [11,12].

Until the present study, to our knowledge, there were no reported attempts to apply human DR-ELISA for the detection of anti-SARS-CoV-2 antibodies in animals. The purchase of diagnostic kits for the detection of SARS-CoV-2 in different biological species—humans on the one hand and animals on the other—complicates the diagnostic process, the supply and management of laboratories, and, consequently, the complex attempts to combat the virus. Therefore, the aim of this study was to verify whether DR-ELISA for the detection of human anti-SARS-CoV-2 antibodies can be applied in cats. In addition, we analyzed the influence of some risk factors on the spread of the virus in the cat population in Bulgaria.

## 2. Materials and Methods

### 2.1. Samples

This study included the following 92 serum samples from cats: 24 stray cats (which live and survive on their own) and 68 domestic cats (pets), obtained in the period of January–June 2021. The samples were obtained from the following three regions of Bulgaria: Veliko Tarnovo (North-Central Bulgaria), Sliven and Stara Zagora (South-Eastern Bulgaria), or between N 43.173512 and N 42.253300, E 25.861241 and E 25.158116. The tested cats were clinically healthy. They were 3 months to 20 years of age (6.57 ± 5.19/mean ± SD; interquartile range, IQR: 6.25); 42 males and 50 females. Samples were collected from cats with indoor and outdoor habitats and from three places of inhabitance. The sera were aliquoted into 100 μL vials and were stored at −80 °C until testing.

A total of 17 pre-pandemic sera (kindly provided by Dr. Todorov, G-Lab, Ltd.) were tested. Among them, there were eleven negatives for *Feline coronavirus* (FcoV), namely, the following: Lab. Code: 70B/2017 positive for *Feline herpesvirus* (FHV), *Feline calicivirus* (FCV); 95B/2017 positive for *Feline immunodeficiency virus* (FIV) and *Feline leukemia virus* (FeLV); 270B/2017 positive for FHV and *Chlamydophila felis*; 137B/2017; 139B/2017; 141B/2017; 23B/2019; 24B/2019; 25B/2019; 30B/2019 and 33B/2019. The serum samples positive for FcoV (n = 6) were 183B/2017; 259B/2017; 260B/2017; 297B/2017; 298B/2017 and 301B/2017. The samples were confirmed via a commercial immunochromatographic FcoV Ab Test Kit (BioNote, Hwaseong-si, Gyeonggi-do, Republic of Korea). The same samples were used to test the cross-reactivity of the two SARS-CoV-2 ELISA kits included in this study. 

The questionnaire asked the owners the following: “Have you had SARS-CoV-2 in the last 2 months; does your cat have outdoor access or does it live inside only?”

### 2.2. Serological Study

To detect antibodies against SARS-CoV-2 in cats, we used a human DR-ELISA for IgA, IgG and IgM antibodies, in which the conjugate is a peroxidase-labeled SARS-CoV-2 N protein (Ingezim COVID 19 DR, Eurofin, Spain). The comparison between Indirect ELISA and DR-ELISA inspired by [13] is shown in Figure 1.

Prior to the reactions, the sera were heat inactivated at 56 °C for 30 min.

The procedure was performed according to the manufacturer’s instructions.

Validation of results: OD of positive control (PC) > 0.5 and PC/NC (negative control) > 2; 

Cut-off point calculation: positive cut-off point = S/P = 6; negative cut-off point = S/P = 4; doubtful between 4 and 6;

Interpretation of the results: The S/P of each sample was calculated as follows: 

S/P = ((OD _sample_ − OD _NC_)/(OD _PC_ − OD _NC_)) × 10

The results were read using an ELISA Microplate Reader, BioBase (China).

To confirm the DR-ELISA results, we also tested all samples with a validated commercial assay for detection of anti-SARS-CoV-2 antibodies, ID Screen SARS-CoV-2 double antigen multi-species ELISA (Innovative Diagnostics, Grabels, France), according to the manufacturer’s instructions. The sensitivity and specificity of the two ELISA kits were determined as follows [14]:

*sensitivity* = (*a*/*a* + *c*) × 100;

*specificity* = (*d*/*d* + *b*) × 100;

*general agreement between the two ELISA kits* = (*a* + *d*/*N*) × 100, where:

*a* is the number of samples that were positive in the two assays;

*d* is the number of samples that were negative in the two assays;

*b* is the number of samples that were positive in the new assay but negative in the standard assay;

*c* is the number of samples that were negative in the new assay but positive in the standard assay;

*N* is the sum of *a* + *b* + *c* + *d*.

### 2.3. RNA Extraction and RT-PCR

RNA was extracted from the ELISA-negative serum samples (to detect pre-symptomatic cats) using an ISOLATE II RNA Mini kit (Bioline, Meridian Bioscience, Memphis, TN, USA) according to the manufacturer’s instructions. RT-PCR was run as described by [7].

### 2.4. Statistical Analysis

To assess the statistical significance of the differences in the mean values of quantitative parameters between ELISA-negative and ELISA-positive animals, we used a one-way analysis of variance (ANOVA). For the categorical parameters, we used the *Chi*-square test. Age was considered a dichotomous variable, with the cut-off value determined using ROC and Youden index analysis. 

The effect of each of the studied parameters as an individual risk factor for the presence of antibodies against SARS-CoV-2 was examined by logistic regression. The odds ratio (OR) and its 95% confidence interval were calculated for each parameter to quantify the association between the risk factor and SARS-CoV-2 seroprevalence. The statistical analysis was performed using MedCalc v. 10.2.0.0 (Ostend, Belgium). 

## 3. Results

The samples were tested using DR-ELISA in the following two independent runs and in duplicate: there were two replicate wells in the first run and single wells in the second run; the results from the two runs showed no differences. Each run gave PC/NC values of 1.135/0.279 OD and 0.961/0.275 OD, respectively. The OD values of the positive samples in the two runs varied between 1.073–3.893 and 0.890–1.571, respectively. The highest values (above the PC) were measured among the stray cats, from 1.185 to 3.893. Two samples in the first run and one in the second run gave OD values of 0.645 (domestic), 0.656 (stray) and 0.563 (domestic), respectively, which classified them as borderline. 

The pre-pandemic samples tested negative in our study with OD values of 0.130–0.196 (which was close to but below the negative control in the kit). These values were similar to those of other negative pandemic samples. Neither of the two ELISA kits showed cross-reactivity with any of the pre-pandemic samples. The validated commercial ELISA for the detection of anti-SARS-CoV-2 Ab in animals confirmed the results from the human ELISA, with 100% specificity and 100% sensitivity between the two assays.

The seroprevalence analysis of pandemic sera showed that a total of 48/92 (52.17%) tested samples were positive for antibodies against SARS-CoV-2 (Table 1). Of these positive samples, 20 were from stray cats, accounting for 21.74% of the total number of tested samples (20/92); 28 were from domestic cats, i.e., 30.43% (28/92). Within each group, the positives were 20/24 (83.33%) among the stray cats and 28/68 (41.18%) among the domestic cats. The positive cases in the regional cities were 42.86%, in small towns 50% and in villages 78.26%.

In the questionnaire, only nine owners had reported positive test results confirming a SARS-CoV-2 infection. The others had not had symptoms of the disease and therefore had not been tested. Since some of these owners’ cats had antibodies against the virus but lived indoors only, it is likely that their owners had been asymptomatic SARS-CoV-2 carriers.

The statistical analysis (logistic regression) of the risk factors for the spread of SARS-CoV-2 among the tested cats in Bulgaria (Table 2) showed that the statistically significant risk factors were age and habitat. The analysis outlined an age limit of 7 years (*p* = 0.001), with cats under 7 years having a five times higher risk than those over 7 years (Figure 2). In terms of habitat, the risk was seven times higher for stray cats than for domestic cats (*p* = 0.001), and in terms of the place of inhabitance, the risk was the highest for cats raised in villages (*p* = 0.006).

When we tested the ELISA-negative samples using RT-PCR, none were positive for SARS-CoV-2 in this assay.

## 4. Discussion

ELISA is a method for the detection of antibodies that is based on the structure and properties of antibodies as follows: a species-specific Fc region and a variable region with two ends that targets a very specific antigenic epitope. Most SARS-CoV-2 kits under development [15] and those commercially available are of the indirect type. They detect specific human anti-SARS-CoV-2 antibodies via a conjugate with labeled anti-human antibodies targeting the Fc region. In dual recognition (DR) (or double antigen) ELISA, the conjugate is a labeled antigen identical to that loaded on the plate, which binds to the free arm of the variable region of the antibody bound to the loaded antigen [13] (Figure 1).

In such studies, the presence of cross-reactions with closely related viruses is important. In this case, the question is whether there are such reactions with FcoV [15]. In the study of Yilmaz et al. [16], antibodies against SARS-CoV-2 were found in 6 out of 34 cats before the pandemic. The authors point out that a possible reason for their positive test results may be the circulation of a SARS-CoV-2-like virus in cats. However, the number of pre-pandemic samples from cats tested in different studies is limited, suggesting various interpretations. Our results and those of the IDV France company (performed for validation of the kit described in Materials and Methods) did not show evidence of the circulation of a SARS-CoV-2-like virus in cats. Some other possible reasons for the positive test results, according to the authors, are cross-reactions with FcoV based on the N protein of both viruses. However, the results of the authors might be due to a technical inaccuracy in conducting the reactions or different ELISA techniques and antigens that they used. Our study showed compelling evidence that cross-reactions with FcoV did not occur in the assays that we used. The evidence in support of the lack of such cross-reactions is the genetic difference between the two viruses [17] demonstrated by phylogenetic analysis of the spike (S), envelope (E), membrane (M) and nucleocapside (N) proteins [17], the different receptors they use [18] and the serological studies carried out so far by means of DR ELISA based on the S protein [8,19,20] and the N protein of SARS-CoV-2 [21,22]. 

Unlike Zhao et al. [23], who developed an indirect ELISA for the detection of antibodies against the SARS-CoV-2 N protein, the commercial kits that we used did not show cross-reactivity. A possible explanation could lie in the different types of ELISA (Figure 1). This could concern some specifics of the antibodies and the FCoV and SARS-CoV-2 antigens that they are against (e.g., virus strains, antigen preparation and epitope characteristics) [24,25].

Testing the SARS-CoV-2-positive samples via an FCoV test cannot distinguish between feline exposure to FCoV and cross-reactivity with anti-SARS-CoV-2 antibodies. That is why we used a validated commercial assay to confirm our results. The commercial assay confirmed our data because the two ELISA kits share a common principle for the detection of antibodies. 

It is possible for cross-reactivity to occur with positive FCoV sera in the case of Ab against the S protein, owing to the AA sequence identity between the S protein of these two viruses, the S protein cleavage site [17] and S2 [26]. With an 85% homology between the human and feline ACE2 receptors [23], this allows the two viruses to infect both hosts [17,27].

The obtained OD values of PC and NC validated the reactions, allowing interpretation of the results. High OD values correspond to a greater amount of Ab, which in stray cats may result from recent infection or continuous contact with the virus stimulating the production of antibodies. However, it is also possible for SARS-CoV-2 infections to stimulate T cells only, without activation of B cells [28].

In the cases of domestic and stray cats with borderline samples, the OD values (indicating the presence of specific anti-N Ab) gave a borderline result between 4 and 6. This, together with the specificity of the reaction, the faster time-dependent decrease in anti-N compared to anti-S SARS-CoV-2 antibodies [29] and the information from the owners (who had had contact with the virus), led us to interpret the results as positive.

At the time of sample collection, we surveyed the owners about prior SARS-CoV-2 infections detected by either RT-PCR or rapid antigen test. The majority of owners were asymptomatic; only nine had had positive test results (confirmed via a laboratory test). However, the positive cases we found in this group of cats are in agreement with other reports that clearly associate SARS-CoV-2 positive cases in domestic cats with the presence of infection in the owners [11,21,30,31]. Our results also showed that the sex of the cats had no relation to the transmission of the virus among them (*p* > 0.05), which was in agreement with other reports [32].

In domestic cats, the risk of contracting the virus was lower because there is a direct association with the owner(s)’ SARS-CoV-2 health status. In stray cats, the risk was seven times greater because exposure to the virus can come from the following different sources: contact with other cats and surfaces contaminated with the virus or due to the spread of infectious aerosol and material. Since the infectious aerosol from a person sneezing can spread to a distance of 6–8 m [33], this aerosol can concentrate in the spatial habitat of the cat and cause infection without immediate human contact, as with domestic cats. This process depends on the persistence of the virus in the environment, which is related not only to the aerosol particle size [33] but also to temperature and sunlight [34]. When including these parameters, ref. [34] found that at 10 °C and 40 °C at 20% relative humidity, the virus loses 90% of its infectivity, respectively, in 10.9–4.7 min, at 40° north latitude. With an increase in humidity, the virus inactivation time increases to 58 min. This indicates that it is possible for the virus to survive and spread through aerosol, dust particles and feces on various surfaces that cats come into contact with in the environment.

Other studies have focused on cats that have contact with the outdoor environment but have owners [35] or on cats that are abandoned or in rescue shelters [8], where they come into contact with the animal keepers. Unlike these reports, we observed a positive seroprevalence among stray cats. Cats with active infection can emit the virus [2,5,7,10,11]. In addition, another study detected the virus in stray dogs from the Amazon rainforest [36]. This accumulating evidence suggests that an infection can occur in susceptible species independently from human contact, possibly via animal-to-animal transmission in accordance with SARS-CoV-2 ecology and biology. 

There is no domestic cat registry in Bulgaria, but there is a program to monitor the number of stray cats in some cities, V. Tarnovo (included in this study), Sofia and Plovdiv (Four Paws Foundation 2016, Municipality of Plovdiv, 2020) [37]. However, the final results are not yet available for discussion. There are just local results that could serve as a basis for a rough approximation. For example, in one neighborhood of the city of Sofia (P. Yavorov, Sredets region) with a population of 31,649 (grao.bg) [38] and a territory of 3 km^2^, there were 454 stray cats (Four Paws Foundation 2016) [37]. Our results in the studied period, the first half of 2021, suggest that despite the likely large number of stray cats in large cities, the risk of transmission and spread of the virus among them was low. Partly this could have resulted from the introduction of restrictive measures that prevented the free movement of people, and thus the possibility for the virus to spread in the environment. In addition, control of the compliance with the anti-epidemic measures in Bulgaria was mainly enforced in large cities.

All these factors together contributed to a higher risk of infection in cats in villages (OR = 4.8000) than in cities. This is because it is common for domestic cats in Bulgarian villages to be allowed to roam free in a manner similar to stray cats. 

We also suggest that, since young cats are more active, they have a greater risk of infection associated with life in the streets or in villages. However, age and sex were not relevant in the epizootology of SARS-CoV-2 in domestic cats in our study. In contrast, [35] found a higher percentage of seropositive cats between 4 and 7 years of age (33.36%) compared to those over 8 years (16.67%). These differences may be due to different times of sampling and contact with the virus, different periods for the decay of antibodies [26], the greater activity of these cats and, consequently, a greater possibility of contact with the owners and infected surfaces.

The positive samples in our study accounted for higher percentages (stray—83.33%, domestic—41.18%, in large cities—42.86%, in small towns—50% and in villages—78.26%) than in other reports as follows: 14.7% in Wuhan [8], 21.7% [35] and 5% [21] in Portugal, Istanbul 28.38% [16], 5.8% in Italy [30], Germany 0.69% [19] and France 21–53% [20]. These are countries that enforced very strict measures to combat SARS -CoV-2. Hence, the differences in the reported seroprevalence could stem from the unknown large number of cats and their greater possibility of contact with the virus for the above reasons. The seroprevalence may be even higher since antibodies against the S protein persist longer in the body compared to those against the N protein [29].

Our results are in contrast to another report [39], which observed no differences in the positive samples in stray cats during different waves of the COVID-19 outbreak in the city of Zaragoza, Spain, for the period of October 2020—January 2022. These contrasting results could be attributed to the different origins of the samples (place of inhabitance, country), hence, different degrees of adherence to the anti-epidemic measures and the consequences thereof. The experimental data have shown that the droplets and aerosols generated from human expiratory activities have a velocity of 1 m/s to 50 m/s [40,41,42] depending on the mechanism—exhaled air, coughing or sneezing– as reviewed in detail in [33]. The route of the SARS-CoV-2-laden human aerosols in the environment shows that cats are at risk of infection due to the aerosol dispersion within their living space (up to 30–40 cm from the ground level). These data, along with the high percentage of asymptomatic cases or unwillingness to get tested because of the ensuing quarantine measures, climatic features and the movement of people in the country [43], explain the high percentage of positive cases in cats in Bulgaria. The high seroprevalence in cats might reflect the two peaks of COVID-19 in humans relevant to the period of our study (January–June 2021), namely, 23 November–11 December 2020 and 21–31 March 2021, with the positives varying in the range of 38.8–40.6% and 22.8–23.5%, respectively [44]. Increases in the positive cases in domestic cats during waves of human infection correlate with the positive cases in the owners [45].

Thus, we speculate that the seroprevalence in street/stray cats could serve as an indicator of the degree of adherence to the anti-epidemic measures in a region/territory, city or country. Cats might be used as sentinel animals to track the virus in nature and in inhabited areas (strays) and to detect asymptomatic cases in humans/owners.

## 5. Conclusions

The present study demonstrates that dual recognition ELISA designed for the control of SARS-CoV-2 in humans is suitable for the study of SARS-CoV-2 in the feline population and other species susceptible to this virus. To the best of our knowledge, this is the first serological study on the spread of SARS-CoV-2 in cats in Bulgaria that confirms not only the human-to-cat transmission of the virus but also the human-independent spread among cats, subject to the natural circulation and transmission of the virus. This, along with non-compliance with anti-epidemic measures, can lead to a higher risk of infection in stray and village cats. The results obtained suggest that cats may serve as an indicator of the degree of compliance with anti-epidemic measures against SARS-CoV-2 in humans in a particular region/territory, city or country.

## Figures and Tables

**Figure 1 vetsci-10-00042-f001:**
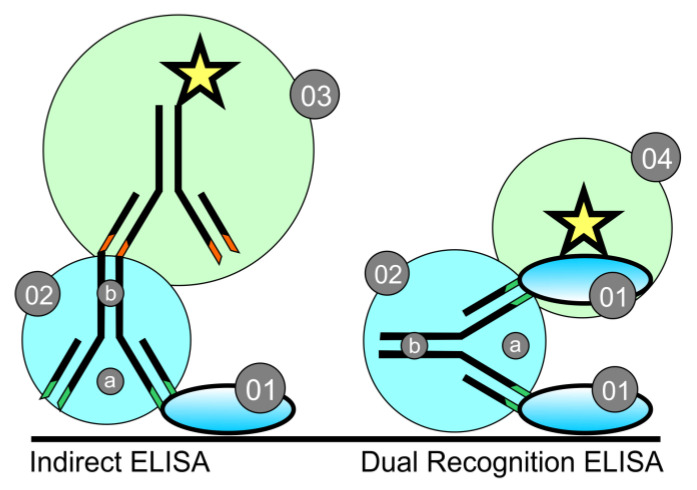
Schematic comparison between indirect ELISA (left) and dual recognition ELISA (right) (inspired by [13]). 01—fixed antigen (Ag); 02—specific antibody (Ab): (a) variable region of the specific Ab; (b) Fc—species-specific region of the Ab; 03—conjugate with labeled (an enzyme represented as a star) secondary Ab for indirect ELISA; 04–conjugate with labeled Ag for dual recognition ELISA.

**Figure 2 vetsci-10-00042-f002:**
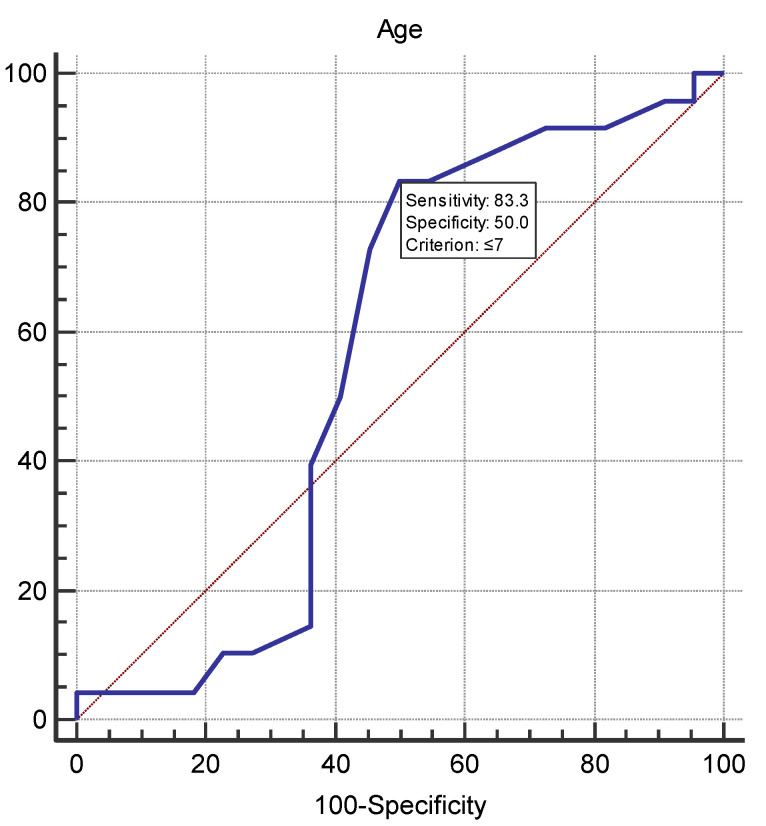
ROC curve for age as a continuous variable. The age cut-off value was ≤7 with area under the curve, sensitivity and specificity of 0.566, 83.3% and 50%, respectively. The dashed diagonal line corresponds to predictive value no better than chance.

**Table 1 vetsci-10-00042-t001:** SARS-CoV-2 seroprevalence in 92 serum samples obtained from domestic and stray cats according to sex, age and place of inhabitance.

Parameter		ELISA Neg (n = 44) Mean ± SD ^1^ or Number (%)	ELISA Pos (n = 48) Mean ± SD ^1^ or Number (%)	*p*-Value
Sex	Male	18 (41)	24 (50)	0.506
	Female	26 (59)	24 (50)	
Age, years		6.3 ± 4.8	5.8 ± 4.1	0.601
Age	>7 years	24 (55)	40 (83)	0.006
	≤7 years	20 (45)	7 (17)	
Habitat	Indoors	40 (91)	28 (58)	0.001
	Outdoors	4 (9)	20 (42)	
Place of inhabitance	Regional city	36 (82)	27 (56)	0.014
	Small town	3 (7)	3 (6)	
	Village	5 (11)	18 (38)	

^1^ SD, standard deviation.

**Table 2 vetsci-10-00042-t002:** Statistical analysis of risk factors for the spread of SARS-CoV-2.

Variable		Odds Ratio	95% CI	*p*-Value
Age	>7 years	−		0.001
	≤7 years	5.000	1.91–13.09	
Habitat	Indoors (domestic)	−		
	Outdoors (stray)	7.1429	2.2011–23.1797	0.001
Place of inhabitance	Regional city	−		
	Small town	1.3333	0.2494–7.1278	0.737
	Village	4.8000	1.5829–14.5560	0.006

## Data Availability

The data presented in this study can be found in the manuscript.

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
