# Peer review of "Human ELISA Detects anti-SARS-CoV-2 Antibodies in Cats: Seroprevalence and Risk Factors for Virus Spread in Domestic and Stray Cats in Bulgaria"

_vetsci, 2023, doi:10.3390/vetsci10010042_

Round 1
Reviewer 1 Report (Previous Reviewer 2)
The manuscript is improved by adding new analysis to confirm the initial results. I have only minor suggestions as follow.
Title
In the title I suggest to add “in Bulgaria”
Abstract
Line 33-34: “In addition, the results indicate the highest risk for cats in villages (p=0.006).” compared to cats in?
M&M
Please add more information on cats evaluated in this study. In addition, also the origin of the feline sera is not properly indicated.
In cats evaluated in this study, blood samples were specifically drawn for the study. Did the Authors ask for an ethical committee approval?
Informed Consent Statement: Informed consent was obtained from all subjects involved in the study. Who are all the subject involved in this study???
Institutional Review Board Statement: Not applicable. – not agree
Line 78: “The tested cats were clinically healthy.“ - How did the authors evaluate the health status of all cats?
Line 78-79: “They were 3 months to 20 years of age” - please add SD or IRQ based on distribution of data
Line 79-80: “The sera were aliquoted into 100 μl vials and were stored at -80 °Ð¡ until testing.“- please add when blood drawn were performed and delete the underline under aliquoted
Line 86: correct FcoV with FCoV
Line 125: “RNA Extraction and RT-PCR” - please state before in the manuscript the reason to perform this test
Line 136: here the authors state that they used OR, than they reported the Risk ratio (table 2) and in the discussion they mention RR (relative risk, line 271). These statistical parameters are different, please be consistent with the one the authors evaluated in their statistical analysis.
Line 152-154: “The validated commercial 152 ELISA for detection of anti-SARS-CoV-2 Ab in animals confirmed the results from the Human ELISA, with 100% specificity and 100% sensitivity between the two assays.” - This data seems not to be realistic. At least a CI should be given.
Table 1 is hard to read, please correct the order of the lines
Figure 2 is hard to understand, how can we understand that the empty circle corresponds to a cut-off<7 years?
Discussion
Line 197-208: In a study that was published in 2019 it was shown that several pre-COVID samples and samples from FCoV type specific cat sera reacted with the N protein of SARS-CoV-2 (Zhao et al, DOI: https://doi.org/10.3201/eid2705.204055). Please comment also the results of this study in your discussion here
Line 197: “In the study of [16] antibodies against”, please use the name of the authors here in addition to the reference number
Line 198: “SARS-CoV-220 were found in 6 out of 30 cats before the pandemic.” Please correct the number after SARS-CoV
Line 233-234: “Our results also showed that the sex of the 233 cats had no relation to the transmission of the virus among them (p >0.05).” - please add some reference of previous studies in which sex was explored but did not resulted a risk factor
References
In the references there is two (n.9 and 27) which is a MedXRiv of 2020, then a preprint which did not get published after 2 years, it should not be cited; or otherwise please add the full text published after the preprint
Author Response
We thank the reviewers for their constructive comments, questions and recommendations. We have revised the manuscript according to them. We hope the reviewers and editors find the revised manuscript considerably improved and suitable for the journal.
Reviewer 1
Title
In the title I suggest to add “in Bulgaria”
Response: Done.
Abstract
Line 33-34: “In addition, the results indicate the highest risk for cats in villages (p=0.006).” compared to cats in?
Response: Done: “In addition, the results indicate that the risk was the highest for the cats in villages (p=0.006)” as compared to cats in other places of inhabitance.
M&M
Please add more information on cats evaluated in this study. In addition, also the origin of the feline sera is not properly indicated.
Response: Done as follows: Samples were collected from cats in an indoor or outdoor habitat and from three places of inhabitance.
In cats evaluated in this study, blood samples were specifically drawn for the study. Did the Authors ask for an ethical committee approval?
Response: Yes. The Ethics Committee approval is an integral part of the procedure for research project funding approval.
Informed Consent Statement: Informed consent was obtained from all subjects involved in the study. Who are all the subject involved in this study???
Response: The statement refers to the cats’ owners. The sentence was polished accordingly: “Informed consent was obtained from all owners of the cats included in the study.“
Institutional Review Board Statement: Not applicable. – not agree
Response: This research project was approved by the Review Board of the Medical University of Sofia, which includes several commissions (Scientific, Ethics and Administrative). The project was granted approval (no. D-80/04.06.2021); hence, performing the study and publishing the results are legitimate.
Line 78: “The tested cats were clinically healthy.“ - How did the authors evaluate the health status of all cats?
Response: The main clinical parameters were within the reference range. There were no clinical signs regarding the respiratory or gastrointestinal tract.
Line 78-79: “They were 3 months to 20 years of age” - please add SD or IRQ based on distribution of data
Response: age: 6.57 ± 5.19 (mean ± SD); interquartile range (IQR): 6.25
Line 79-80: “The sera were aliquoted into 100 μl vials and were stored at -80 °Ð¡ until testing.“- please add when blood drawn were performed and delete the underline under aliquoted
Response: Blood samples were drawn during the period of the study described in Materials and methods The underline was deleted.
Line 86: correct FcoV with FCoV
Response: Done.
Line 125: “RNA Extraction and RT-PCR” - please state before in the manuscript the reason to perform this test
Response: This was done as per another peer-reviewer’s recommendations. Their recommendation and our response follow below:
“Also, I see that your group developed RT-PCR [7]. It would be interesting to see how many of your positive samples were also PCR-positive.”
Response: We do not have clinical samples from these cats; all were clinically healthy. This has been added in the text. Comparing ELISA results with RT-PCR results is not completely accurate because SARS-CoV-2 infection does not always lead to Ab production; in some cases, it induces just cellular immunity. This is discussed in another paper of ours. Detection of antibodies in the serum or plasma indicates absence of free virus particles there. That is why we tested the negative samples via RT-PCR to detect potential pre-symptomatic cases.
Line 136: here the authors state that they used OR, than they reported the Risk ratio (table 2) and in the discussion they mention RR (relative risk, line 271). These statistical parameters are different, please be consistent with the one the authors evaluated in their statistical analysis.
Response: The correct term odds ratio (OD) is now used throughout the text.
Line 152-154: “The validated commercial 152 ELISA for detection of anti-SARS-CoV-2 Ab in animals confirmed the results from the Human ELISA, with 100% specificity and 100% sensitivity between the two assays.” - This data seems not to be realistic. At least a CI should be given.
Response: These two tests are identical in terms of antigen and underlying principle, but are commercialized under different trademarks and registration (for human and veterinary use). Such confirmatory analysis was recommended by another peer-reviewer to confirm the results. We did not find it surprising that the results were 100% confirmed because the two assays have the same principle but different packaging, trademarks and registration (for human and veterinary use).
Table 1 is hard to read, please correct the order of the lines
Response: Table 1 was redesigned to become more comprehensible.
Figure 2 is hard to understand, how can we understand that the empty circle corresponds to a cut-off<7 years?
Response: Figure 2 was replaced with an original graph provided by the software with information for the variable “Age”.
Discussion
Line 197-208: In a study that was published in 2019 it was shown that several pre-COVID samples and samples from FCoV type specific cat sera reacted with the N protein of SARS-CoV-2 (Zhao et al, DOI: https://doi.org/10.3201/eid2705.204055). Please comment also the results of this study in your discussion here
Response: According to this recommendation, the following text was added: “Unlike Zhao et al., who developed an indirect ELISA for detection of antibodies against the SARS-CoV-2 N protein, the commercial kits that we used did not show cross-reactivity. A possible explanation could lie in the different types of ELISA (Fig. 1). This could concern some specifics of the antibodies and the FCoV and SARS-CoV-2 antigens that they are against (e.g. virus strains, antigen preparation and epitope characteristics) (Shrock et al., 2020, Crill et al., 2004)“.
Line 197: “In the study of [16] antibodies against”, please use the name of the authors here in addition to the reference number
Response: We used the name of the authors.
Line 198: “SARS-CoV-220 were found in 6 out of 30 cats before the pandemic.” Please correct the number after SARS-CoV
Response: The technical mistakes were revised accordingly.
Line 233-234: “Our results also showed that the sex of the 233 cats had no relation to the transmission of the virus among them (p >0.05).” - please add some reference of previous studies in which sex was explored but did not resulted a risk factor
Response: A reference was added: “Our results also showed that the sex of the cats had no relation to the transmission of the virus among them (p >0.05), which was in agreement with other reports (Westman et al.2016).”
References
In the references there is two (n.9 and 27) which is a MedXRiv of 2020, then a preprint which did not get published after 2 years, it should not be cited; or otherwise please add the full text published after the preprint
Response: We agree. Since every journal has specific policies and requirements that apply to the citation of pre-print versions, it is very helpful when they are specified in the instructions for authors.
Reference 27 has a published version and this has been updated.
Reviewer 2 Report (Previous Reviewer 3)

Author Response
We thank the reviewers for their constructive comments, questions and recommendations. We have revised the manuscript according to them. We hope the reviewers and editors find the revised manuscript considerably improved and suitable for the journal.
Reviewer 2
We are thankful for the questions, comments and recommendations.
Introduction:
Line 42: I still disagree with your wording here as I mentioned in the previous version. The best way
to state this is to say “Animal species known to be highly susceptible include….”
Because you only named 3 species and there are many more species that are highly susceptible.
Response: According to this recommendation, the sentence was revised as follows: “Animal species known to be highly susceptible include cats, ferrets [2] and minks [4]”
Line 44: “as the animal models” does not make sense at the end of that sentence.
Figure 1 is still not clear to me. What is the purpose of the arrows pointing to the labels? As I said
previously, a better explanation of the figure belongs in the caption.
Response: The phrase “as the animal models” was deleted. The Figure was revised as recommended.
Material & Methods:
What dilution of serum did you use in your assays and how was this determined?
Response: “The procedure was performed according to the manufacturer's instructions“, as described in Materials and methods.
The manufacturer’s instructions are as follows: “Make a 1/5 dilution in diluent DE13. This dilution can be performed directly in the well by first adding the diluent and then the sample (40ul of diluent and 10ul of serum).“
Discussion:
Line 196-197: Be more concise. “One and the same” is a redundant phrase.
FCoV and SARS-CoV-2 are in the same family (Coronavidae) but they are in different genera and are genetically distinct from one another according to one of your references.
Response: The redundant phrase “because they belong to one and the same family“ was deleted.
Line 198: says SARS-CoV-220.
Also, the reference you cited (16) says that 6 of 34 seropositive cats were sampled before the emergence of COVID-19 in humans (Dec 2019) and the reasons cited were a cross-reactivity of immune responses between FCoV and SARS-CoV-2, most likely to the nucleocapsid protein; or (ii) a SARS-CoV-2-like virus circulating in cats in Istanbul.
Response: Тhe technical errors were corrected.
There have been speculations about possible pre-pandemic circulation of a SARS-CoV-2-like virus in Istanbul. If there is such possibility, then it should apply to Bulgaria, too, for geographical and ethnic reasons associated with continual movement of people and goods. Moreover, there are very few pre-pandemic samples. For example, even the IDV France company used just 16 pre-pandemic samples to verify their commercial kit (we tested 17 ones). This is why it is difficult to draw final conclusions and many speculations are possible. When testing our pre-pandemic samples, we found no cross-reactions or SARS-CoV-2-like virus.
The following text was added: “The authors point out that a possible reason for their positive test results may be the circulation of a SARS-CoV-2-like virus in cats. However, the number of pre-pandemic samples from cats tested in different studies is limited, suggesting various interpretations. Our results and those of the IDV France company (performed for validation of the kit described in MM) did not show evidence of circulation of a SARS-CoV-2-like virus in cats. Some other possible reasons for the positive test results, according to the authors, are…“
199-200: “given 199 that the authors were developing indirect ELISA based on the S protein and the
RBD region of the S protein.” This should be deleted, as it is irrelevant and confusing to the point you are trying to make in this paragraph.
Response: This text was deleted.
201-202: “However, it is likely that this result was due to a technical inaccuracy in conducting the reactions, e.g. incomplete homogenization, washing, etc.” What result are you referring to here?
Response: The cross reactivity between SARS-CoV-2 and FCoV. After the recommended deletion of the text from the previous sentence, it is clear to which result it refers to.
202-203: “Studies have shown compelling evidence that cross-reactions with FCoV are not possible in these methods.” What methods are you referring to here?
This paragraph should be completely rewritten.
Response: The whole paragraph from line 199 to 203 was revised.
Lines 209-210: “Testing the SARS-CoV-2-positive samples via an FCoV test cannot distinguish between feline exposure to FCoV and cross-reactivity with anti-SARS-CoV-2 antibodies.”
How do you know this?
Response: To detect cross-reactivity with FCoV, we used the pre-pandemic samples described in MM. No cross-reactions were detected. FCoV circulates in Bulgaria and is a common cause of infections (unpublished data). That is why positive results with the FCoV assay used by us will show its presence in the tested samples, which was not the purpose of this study. Another possibility that cannot be excluded is the potential exposure to both viruses and production of antibodies against them.
Line 214-217:Probably means that there is a high chance that something will happen. Possible means that something is not impossible.
You should not use these two words together in the same sentence. You contradict yourself here with what you say previously about cross reactivity. The spike protein is the most hypervariable of coronavirus proteins and the least likely to be cross reactive with other coronaviruses as you mentioned previously. “With 85% homology between the human and feline ACE2 receptors [23], this allows the two viruses to infect both hosts [17].” Are you saying here that FCoV can infect humans? Please reword this sentence to be more clear.
Response: The claim that FCoV can infect humans is not ours. “However, cats have been shown to be susceptible to SARS-CoV-2, and FCoV also had been shown to infect human”. That is why we added the reference accordingly: “With 85% homology between the human and feline ACE2 receptors [23], this allows the two viruses to infect both hosts [17, 27].”
Line 224: Didn’t you say previously that borderline OD values were between 0.4 and 0.6?
Response: According to the manufacturer’s instructions, the doubtful S/P values are between 4 and 6. The values of 0.4 and 0.6 are the result for OD of samples before multiplication by 10 as pointed out in the instructions.
The manuscript was polished by a professional English editor.
This manuscript is a resubmission of an earlier submission. The following is a list of the peer review reports and author responses from that submission.
Round 1
Reviewer 1 Report
The piece of writing is, by and large, well written. It is also worth mentioning the novelty of the study,
since it turns out that no other prior seroprevalence essay has been carried out in Bulgary so far.
The discussion also gives a better understanding about potential shortcomings in a serosurveillance study.
However, I would like to ask the authors about the negative controls. In line 64 the authors say that the study included 92 samples: 24 stray cats and 68 domestic cats.
However, it is not clear how many of them correspond to negative serum. In addition, negative serum should be pre-pandemic and
this is also not mentioned. please, clarify this point.
Line 92 ROC analysis: it would be necessary to show the ROC graph, as part of results.
Author Response
We thank the reviewers for their constructive comments, questions and recommendations. We have revised the manuscript according to most of them and we provide responses to the rest. We hope the reviewers and editors will find the revised manuscript considerably improved.
- Comments and Suggestions for Authors
The piece of writing is, by and large, well written. It is also worth mentioning the novelty of the study, since it turns out that no other prior seroprevalence essay has been carried out in Bulgary so far.
The discussion also gives a better understanding about potential shortcomings in a serosurveillance study.
However, I would like to ask the authors about the negative controls.
In line 64 the authors say that the study included 92 samples: 24 stray cats and 68 domestic cats.
However, it is not clear how many of them correspond to negative serum. In addition, negative serum should be pre-pandemic and this is also not mentioned. please, clarify this point.
Response: At the beginning of the study [In a preliminary experiment], we included 7 serum samples from 2017 from cats as negative pre-pandemic controls. These samples tested negative in our study. We had not included them in the first version of the manuscript because the laboratory no longer exists –as either a tangible or intangible entity. We have now added these samples and the results in the text, in addition to the 92 samples that were subject to the study reported here.
We added the following sentence in Materials and methods (MM): The tested cats were clinically healthy.“ as well as the following text:
ММ: "We tested seven pre-pandemic sera from 2017 (kindly provided by Dr Todorov, G-Lab, Ltd.), namely: Lab. Code: 70Ð’/2017 positive for Feline herpesvirus (FHV), Feline calicivirus (FCV); 95Ð’/2017 positive for Feline immunodeficiency virus (FIV) and Feline leukemia virus (FeLV); 183Ð’/2017, 259/2017 and 260/2017 positive for Feline coronavirus (FCoV) ; 270Ð’/2017 positive for FHV and Chlamydophila felis; 137Ð’/2017 negative for FcoV.“;
Results: The pre-pandemic samples tested negative in our study with OD values of 0.130 - 0.196 (which was close to but below the negative control in the kit). These values were similar to those of other negative pandemic samples.
Acknowledgement:
"The authors are grateful to Dr Tsveran Todorov (G-Lab, Ltd, Sofia, Bulgaria) for kindly providing pre-pandemic serum samples (from 2017).“
Line 92 ROC analysis: it would be necessary to show the ROC graph, as part of results.
Answer: The ROC curve has been added.
Reviewer 2 Report
The aim of this study was to evaluate if the human DR-ELISA for detection of anti-SARS-CoV-2 antibodies can be applied in cats. The second aim was to investigate risk or protective factors for infection with SARS-CoV-2 among the cat population in Bulgaria. Such a study is interesting in the light of what has been found in the general feline population. In addition, the study is well written and easy to read. Interestingly, the authors found more a very high seroprevalence of SARS-CoV-2 seropositivity respect to studies published far now and a highest seropositivity in stray than in owned cats, differently to what has been documented in the past feline epidemiology studies.
The main concern of this study is the method of SARS-CoV-2 antibodies detection and the reliability of the results obtained with this ELISA. The authors used only one ELISA kit based on the detection of N SARS-CoV-2 antibodies. Possible false positive samples could be identified. As authors explained themselves, cross-antigenic reaction is possible between N protein from FCoV and N from SARS CoV-2. In the study of Zhao et al (DOI: https://doi.org/10.3201/eid2705.204055) it was shown that several pre-COVID samples and samples from FCoV type specific cat sera reacted with the N protein of SARS-CoV-2. Reaction could be also with other coronaviruses eg with canine coronavirus. In addition, the majority of the studies on seroprevalence in cats is based on ELISAs with the S or RBD protein of SARS-CoV-2 and then preferably confirmation with a virus neutralisation assay (with whole virus, pseudotyped virus or surrogate neutralisation assays). Therefore, to further validate this study, I suggest confirming the SARS-CoV-2 antibodies detection either by seroneutralisation (eg a surrogate neutralisation assay), or at least with another ELISA test based on S or RBD protein.
Other comments and minor points.
In the abstract I suggest to avoid the abbreviations (eg mo. instead of months, Jan for January)
Some information on epidemiology of SARS-CoV-2 infection in human beings in Bulgaria in the same time frame of the feline study should be added in the introduction or in the discussion section
The origin of the feline blood samples is not properly indicated. Were these collected specifically for the purpose of this study? Where have been analysed the samples?
The authors should state that the animal study protocol has been approved by an Institutional Review Board and add the protocol number and day-month-year of approvation
Some references are not properly cited in the manuscript (eg ref no. 14 and 13 at lines 142 and 146 respectively
Author Response
We thank the reviewers for their constructive comments, questions and recommendations. We have revised the manuscript according to most of them and we provide responses to the rest. We hope the reviewers and editors will find the revised manuscript considerably improved.
The aim of this study was to evaluate if the human DR-ELISA for detection of anti-SARS-CoV-2 antibodies can be applied in cats. The second aim was to investigate risk or protective factors for infection with SARS-CoV-2 among the cat population in Bulgaria. Such a study is interesting in the light of what has been found in the general feline population. In addition, the study is well written and easy to read. Interestingly, the authors found more a very high seroprevalence of SARS-CoV-2 seropositivity respect to studies published far now and a highest seropositivity in stray than in owned cats, differently to what has been documented in the past feline epidemiology studies.
The main concern of this study is the method of SARS-CoV-2 antibodies detection and the reliability of the results obtained with this ELISA. The authors used only one ELISA kit based on the detection of N SARS-CoV-2 antibodies. Possible false positive samples could be identified. As authors explained themselves, cross-antigenic reaction is possible between N protein from FCoV and N from SARS CoV-2. In the study of Zhao et al (DOI: https://doi.org/10.3201/eid2705.204055) it was shown that several pre-COVID samples and samples from FCoV type specific cat sera reacted with the N protein of SARS-CoV-2. Reaction could be also with other coronaviruses eg with canine coronavirus. In addition, the majority of the studies on seroprevalence in cats is based on ELISAs with the S or RBD protein of SARS-CoV-2 and then preferably confirmation with a virus neutralisation assay (with whole virus, pseudotyped virus or surrogate neutralisation assays). Therefore, to further validate this study, I suggest confirming the SARS-CoV-2 antibodies detection either by seroneutralisation (eg a surrogate neutralisation assay), or at least with another ELISA test based on S or RBD protein.
Response: We agree with all the points made by the reviewer. At the time of purchase of the reagents, there were no specialized kits for detection of animal anti-S SARS-CoV-2 protein Ab (or any other Ab). None have been registered up to now eather. This is actually what prompted us to perform this study.
VNA would not give reliable results in this case because it is based on virus-neutralizing antibodies, which are against the spike glycoprotein (S) of the virus, while the kit we used detects Ab against the N protein, so the results would not be comparable.
Regarding the suggestion to use ELISA for detection of Ab against the S glycoprotein, we had considered using human ELISA (Demeditec COVID-19 quantitative IgG ELISA). However, for the above reasons, and because it is based on the indirect technique, i.e. involving the Fc domain of Ab, this would have required adding an inhibition step with cat serum. This would have modified the method and would have required validation via VNA (detecting the same antibodies).
The reasoning behind our study design was that the ELISA kit we use involves the variable Ab domains, not the Fc domain, and recombinant SARS-CoV-2 N protein.
In addition, at the beginning of the study [In a preliminary experiment], we included 7 serum samples from 2017 from cats as negative pre-pandemic controls. These samples tested negative in our study. We had not included them in the first version of the manuscript because the laboratory no longer exists –as either a tangible or intangible entity. We have now added these samples and the results in the text.
The following text was added:
ММ: "We tested seven pre-pandemic sera from 2017 (kindly provided by Dr Todorov, G-Lab, Ltd.), namely: Lab. Code: 70Ð’/2017 positive for Feline herpesvirus (FHV), Feline calicivirus (FCV); 95Ð’/2017 positive for Feline immunodeficiency virus (FIV) and Feline leukemia virus (FeLV); 183Ð’/2017, 259/2017 and 260/2017 positive for Feline coronavirus (FCoV) ; 270Ð’/2017 positive for FHV and Chlamydophila felis; 137Ð’/2017 negative for FcoV.“;
Results: The pre-pandemic samples tested negative in our study with OD values of 0.130 - 0.196 (which was close to but below the negative control in the kit). These values were similar to those of other negative pre-pandemic samples.
Acknowledgement:
"The authors are grateful to Dr Tsveran Todorov (G-Lab, Ltd, Sofia, Bulgaria) for kindly providing pre-pandemic serum samples (from 2017).“
Other comments and minor points.
In the abstract I suggest to avoid the abbreviations (eg mo. instead of months, Jan for January)
Response: We removed the abbreviations from the abstract.
Some information on epidemiology of SARS-CoV-2 infection in human beings in Bulgaria in the same time frame of the feline study should be added in the introduction or in the discussion section
Response: Epidemiologically, it is of interest to compare the time course in the two species. This requires including a long period of 2 or 3 years. We have submitted such a research proposal for consideration and we are currently awaiting its evaluation.
The following text was added in the discussion: “The high percentage of positive samples in cats might reflect the two peaks of COVID-19 in humans relevant to the period of our study (January – June 2021), namely 23rd November – 11th December 2020 and 21st – 31st March 2021, with the positives varying in the range of 38.8-40.6% and 22.8-23.5%, respectively (https://covid19.ncipd.org/). In addition, the percentage of SARS-CoV-2 positives in the human population could have been higher owing to asymptomatic cases or unwillingness to get tested because of ensuing quarantine measures.
The origin of the feline blood samples is not properly indicated. Were these collected specifically for the purpose of this study? Where have been analysed the samples?
Response: The samples were collected for the purpose of the study and were tested in the Department of Medical Microbiology, Medical University of Sofia. The following sentence was added in ММ:
“The reactions were read using an ELISA Microplate Reader, BioBase (China).“
Part of the samples were collected by co-authors. An acknowledgement was added for the rest: “We thank veterinarians for the sample collection.“
The authors should state that the animal study protocol has been approved by an Institutional Review Board and add the protocol number and day-month-year of approvation
Response: The samples were collected by licensed veterinarians in accordance with the active legislation in Bulgaria and the principles of Good Clinical Practice.
Such information was requested by the editors at an earlier stage and has been provided.
Some references are not properly cited in the manuscript (eg ref no. 14 and 13 at lines 142 and 146 respectively.
Response: Redundant text was removed (lines 139-146)
Author Response
We thank the reviewers for their constructive comments, questions and recommendations. We have revised the manuscript according to most of them and we provide responses to the rest. We hope the reviewers and editors will find the revised manuscript considerably improved.
Sirakov and colleagues present: Human ELISA detects anti-SARS-CoV-2 antibodies in cats:
seroprevalence and risk factors for virus spread in domestic and stray cats
Introduction:
Line 32-33: It is incorrect to claim that the species you mention are the “most susceptible”.
That is not currently defined.What are the parameters for “most susceptible”? Neither of your references (2, 4) make that claim.
Response: We agree that the sentence is incomplete. The following text was added for better clarity: “following experimental infection, as the animal models” and the word “most” was revised to “highly”.
Regarding the parameters, generally they depend on the virus as well as on the host, namely: infectious dose, virus shedding rate, virus replication in various organs, virus detection in different samples, re-isolation, titer etc.
Material & Methods:
2.1 Samples: Not enough information on how the samples were taken and who took the samples.
Response: Part of the samples were collected by co-authors (N.R. and A.R.). An acknowledgement was added for the rest: “We thank veterinarians for the sample collection.“
Also no mention of heat inactivation of the samples before testing.
Serum must be heated to 55 degrees C for a minimum of 30 minutes before testing.
Complement., a component of serum can cause false positives if your samples aren’t heat inactivated.
Response: Although the manufacturer’s instructions do not mention this step (because of the solvents), it is part of Good Laboratory Practices (Sirakov et al. Development of blocking enzyme-linked immunosorbent assay for detection of caprine herpesvirus 1 antibodies in Bulgaria.2016), and as such, is not always explicitly stated. That is why we had not mentioned it. We performed this step at 56°C. The following sentence was added in the text: “Prior to the reactions the sera were heat inactivated at 56°C for 30 min.“
2.2 Serological study: Line 76: The comparison between indirect and Dr-ELISA modified from [13] is shown in Figure 1. Your figure caption says it is modified from [18].
I don’t see anything resembling your Figure 1 in either reference.
Figure 1 is not very clear or helpful. It needs improvement or more of an explanation in the caption.
Response: I agree. I personally made the figure, so the word “modified” was revised to “inspired by” because I got the idea from their figure. With the revisions of the manuscript recommended by you and the other referees, the figure better illustrates the basic idea and the application. It is relevant to the points made by the peer-reviewers regarding VNA and other ELISA techniques.
Results:
Running the sample again does not validate the methods.
Response: We agree. Our explanation was unclear. What we actually meant was that: since in some confirmatory ELISA kits one sample is assayed in two replicate wells, we performed a similar assay to test whether the result would be different from that in a single well, from a practical perspective. The text was revised accordingly: “The samples were assayed in duplicate: there were two replicate wells in the first run, and single wells in the second run; the results from the two runs showed no differences.“
You should be running some confirmatory tests like VNT, which is the gold standard, on your positive samples and some of your negative samples
Response: VNT would not give reliable results in this case because it is based on virus-neutralizing antibodies, which are against the spike glycoprotein (S) of the virus, so they will hinder the virus replication in cell culture and will be read as Ab titer. In contrast, the kit we used detects Ab against the N protein, so the results would not be comparable or accurate. The proposed approach would have been feasible if we were running quantitative ELISA for neutralizing antibodies against SARS-CoV-2.
Also, I see that your group developed RT-PCR [7]. It would be interesting to see how many of your positive samples were also PCR-positive.
Response: We do not have clinical samples from these cats; all were clinically healthy. This has been added in the text. Comparing ELISA results with RT-PCR results is not completely accurate because SARS-CoV-2 infection does not always lead to Ab production; in some cases, it induces just cellular immunity. This is discussed in another paper of ours.
Detection of antibodies in the serum or plasma indicates absence of free virus particles there. That is why we tested the negative samples via RT-PCR to detect potential pre-symptomatic cases.
We added the following text:
MM: “RNA was extracted from the ELISA-negative serum samples using an ISOLATE II RNA Mini kit (Bioline, United Kingdom) according to the manufacturer’s instructions. RT-PCR was run as described by Sirakov et al. 2022.“
Results : “When we tested the ELISA-negative samples using RT-PCR, none was positive for SARS-CoV-2 in this assay.“
Overall, I think using pre-covid serum samples to establish a cut-off would be a more accurate method to determine positivity.
Response: At the beginning of the study [In a preliminary experiment], we included 7 serum samples from 2017 from cats (from the collection of GLab, Ltd, Sofia, Bulgaria) as negative pre-pandemic controls. These samples tested negative in our study. We had not included them in the first version of the manuscript because the laboratory no longer exists –as either a tangible or intangible entity. We have now added these data in the text:
ММ: "We tested seven pre-pandemic sera from 2017 (kindly provided by Dr Todorov, G-Lab, Ltd.), namely: Lab. Code: 70Ð’/2017 positive for Feline herpesvirus (FHV), Feline calicivirus (FCV); 95Ð’/2017 positive for Feline immunodeficiency virus (FIV) and Feline leukemia virus (FeLV); 183Ð’/2017, 259/2017 and 260/2017 positive for Feline coronavirus (FCoV) ; 270Ð’/2017 positive for FHV and Chlamydophila felis; 137Ð’/2017 negative for FcoV.“;
Results: The pre-pandemic samples tested negative in our study with OD values of 0.130 - 0.196 (which was close to but below the negative control in the kit). These values were similar to those of other negative pre-pandemic samples.
Acknowledgement:
"The authors are grateful to Dr Tsveran Todorov (G-Lab, Ltd, Sofia, Bulgaria) for kindly providing pre-pandemic serum samples (from 2017).“
Sincerely,
I.S.
Round 2
Reviewer 1 Report
The manuscript has improved. The authors have clarified the points and added further details.
Reviewer 2 Report
Dear Authors,
thank you for your precise and punctual responses to my concerns. However, in my opinion, without a confirmatory test of the obtained results, these could not be published. The pre-covid serum samples tested are not useful to validate the method used in this survey.
Best Regards
Reviewer 3 Report
The revisions are not adequate.